# Laser excitation of the 1s-hyperfine transition in muonic hydrogen

P. Amaro[1*], A. Adamczak[2], M. Abdou Ahmed[3], L. Affolter[4], F. D. Amaro[5], P. Carvalho[1],
T.-L. Chen[6], L. M. P. Fernandes[5], M. Ferro[1], D. Goeldi[4], T. Graf[3], M. Guerra[1],
T. W. Hänsch[7,8], C. A. O. Henriques[5], Y.-C. Huang[6], P. Indelicato[9], O. Kara[4], K. Kirch[4,10],
A. Knecht[10], F. Kottmann[4,10], Y.-W. Liu[6], J. Machado[1], M. Marszalek[10], R. D. P. Mano[5],
C. M. B. Monteiro[5], F. Nez[9], J. Nuber[4,10], A. Ouf[11], N. Paul[9], R. Pohl[11, 12], E. Rapisarda[10],
J. M. F. dos Santos[5], J. P. Santos[1], P. A. O. C. Silva[5], L. Sinkunaite[10], J.-T. Shy[6],
K. Schuhmann[4], S. Rajamohanan[11], A. Soter[4], L. Sustelo[1], D. Taqqu[4,10], L.-B. Wang[6],
F. Wauters[12], P. Yzombard[9], M. Zeyen[4] and A. Antognini[4,10†]

**1** Laboratory of Instrumentation, Biomedical Engineering and Radiation Physics
(LIBPhys-UNL), Department of Physics, NOVA School of Science and Technology,
NOVA University Lisbon, 2829-516 Caparica, Portugal
**2** Institute of Nuclear Physics, Polish Academy of Sciences, PL-31342 Krakow, Poland
**3** Institut für Strahlwerkzeuge, Universität Stuttgart, 70569 Stuttgart, Germany
**4** Institute for Particle Physics and Astrophysics, ETH Zurich, 8093 Zurich, Switzerland
**5** LIBPhys-UC, Department of Physics, University of Coimbra, P-3004-516 Coimbra, Portugal
**6** Department of Physics, National Tsing Hua University, Hsinchu 30013, Taiwan
**7** Ludwig-Maximilians-Universität, Fakultät für Physik, 80799 Munich, Germany
**8** Max Planck Institute of Quantum Optics, 85748 Garching, Germany
**9** Laboratoire Kastler Brossel, Sorbonne Université, CNRS, ENS-Université PSL,
Collège de France, 75005 Paris, France
**10** Paul Scherrer Institute, 5232 Villigen-PSI, Switzerland
**11** Institut für Physik, QUANTUM, Johannes Gutenberg-Universität Mainz,
55099 Mainz, Germany
**12** PRISMA+ Cluster of Excellence and Institute of Nuclear Physics, Johannes
Gutenberg-Universität Mainz, 55099 Mainz, Germany

★ pdamaro@fct.unl.pt, † aldo.antognini@psi.ch,

## Abstract

The CREMA collaboration is pursuing a measurement of the ground-state hyperfine splitting (HFS) in muonic hydrogen ($\mu$p) with 1 ppm accuracy by means of pulsed laser spectroscopy to determine the two-photon-exchange contribution with $2 \times 10^{-4}$ relative accuracy. In the proposed experiment, the $\mu$p atom undergoes a laser excitation from the singlet hyperfine state to the triplet hyperfine state, then is quenched back to the singlet state by an inelastic collision with a $H_2$ molecule. The resulting increase of kinetic energy after the collisional deexcitation is used as a signature of a successful laser transition between hyperfine states. In this paper, we calculate the combined probability that a $\mu$p atom initially in the singlet hyperfine state undergoes a laser excitation to the triplet state followed by a collisional-induced deexcitation back to the singlet state. This combined probability has been computed using the optical Bloch equations including the inelastic and elastic collisions. Omitting the decoherence effects caused by the laser bandwidth and collisions would overestimate the transition probability by more than a factor of two in the experimental conditions. Moreover, we also account for Doppler effects and provide the matrix element, the saturation fluence, the elastic and inelastic collision rates for the singlet and triplet states, and the resonance linewidth. This calculation thus quantifies one of the key unknowns of the HFS experiment, leading to a precise



definition of the requirements for the laser system and to an optimization of the hydrogen gas target where $\mu$p is formed and the laser spectroscopy will occur.

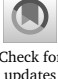

## Contents

## 1 Introduction

Highly accurate measurements of atomic transitions can be used as precise probes of low-energy properties of nuclei. As the Bohr radius of hydrogen-like atoms decreases with increasing orbiting particle mass, muonic atoms (atoms formed by a negative muon and a nucleus) have an enhanced sensitivity to nuclear structure effects [1–7]. The Charge Radius Experiments with Muonic Atoms (CREMA) collaboration in recent years has performed laser spectroscopy of the $2s - 2p$ (Lamb shift) transitions in muonic hydrogen ($\mu$p) [4, 6], muonic deuterium ($\mu$d) [8] and muonic helium ($\mu^4$He$^+$) [9] and extracted the corresponding nuclear

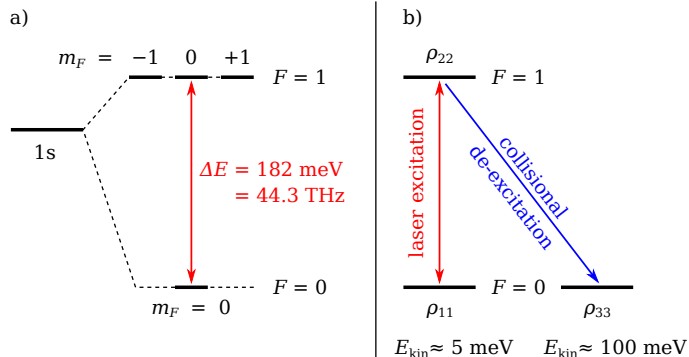

Figure 1: (a) Hyperfine structure of the 1s-state in $\mu$p divided into the triplet ($F = 1$) and the singlet ($F = 0$) states depending on the total angular momentum of the muon-proton system. (b) The three-level system used in the Bloch equations to model the laser excitation followed by collisional deexcitation with an increase of kinetic energy ($E_{\text{kin}}$). Initially all $\mu$p atoms are thermalized (average of $E_{\text{kin}} \approx 5$ meV) to the singlet state with population $\rho_{11}$. The laser pulse drives the HFS transition, exciting the $\mu$p atoms into the triplet state with population $\rho_{22}$. An inelastic collision then deexcites the triplet state back to the singlet state converting the transition energy into kinetic energy. This singlet state with additional kinetic energy is the third level in the optical Bloch equations with population $\rho_{33}$ and $E_{\text{kin}} \approx 100$ meV.

charge radii with an unprecedented accuracy. The impact of the $\mu$p measurements on beyond-standard-model searches, on precision atomic physics, and on the proton structure can be found in recent reviews [7, 10–12]. Along this line of research, the CREMA collaboration is presently aiming at the measurement of the ground-state hyperfine splitting (HFS) in $\mu$p with 1 ppm relative accuracy by means of pulsed laser spectroscopy.

From the measurement of the HFS, precise information about the magnetic structure of the proton can be extracted [13–23]. Specifically, by comparing the measured HFS transition frequency with the corresponding theoretical prediction based on bound-state QED calculations [5, 13, 19, 20], the two-photon-exchange contribution can be extracted with approximately $2 \times 10^{-4}$ relative accuracy. Because the two-photon-exchange contribution can be expressed as the sum of a finite-size (static, elastic) part proportional to the Zemach radius ($R_Z$) and a polarizability part (dynamic, virtual excitation), its determination can be used to extract separately the two parts: the Zemach radius when the polarizability contribution is assumed from theory [13, 15, 16, 18, 21–25], and the polarizability contribution when taking $R_Z$ from electron-proton scattering or hydrogen spectroscopy [19, 26–28].

In this paper, we calculate the laser transition probability between singlet and triplet sublevels of the ground state hyperfine-splitting in $\mu$p (see Figure 1), accounting for the actual detection scheme used in the experiments and considering collisional and Doppler effects. This transition probability is one of the key quantities needed to evaluate the feasibility of the CREMA hyperfine-splitting experiment, and to define the requirements for the experimental setup.

Two other collaborations, the FAMU collaboration at RAL (UK) [20, 29–31] and another one at J-PARC (Japan) [32–34], are also aiming at a precision measurement of the HFS in $\mu$p. For all three experimental collaborations, the main challenge is posed by the small laser-induced transition probability resulting from the small matrix element (magnetic dipole-type transition, $M1$), in conjunction with a transition wavelength at 6.8 $\mu$m where no adequate (sufficiently powerful) laser technologies are available, and the large volume where $\mu$p atoms

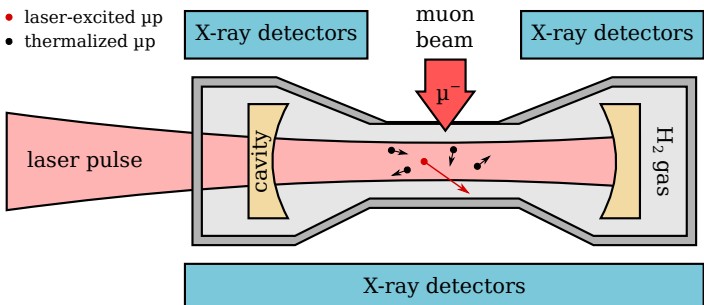

Figure 2: (Color online) Schematic (not to scale) of the setup for measuring the HFS in $\mu$p. A negative muon beam is stopped in a $H_2$ gas target at cryogenic temperatures, pressures of about 1 bar, and a thickness of about 1 mm. The formed $\mu$p atoms are excited by the laser pulse whose intensity is enhanced in the multi-pass cavity. A successful laser excitation of the $\mu$p atoms followed by a collisional deexcitation leads to a $\mu$p atom with extra kinetic energy. With this extra kinetic energy, the $\mu$p atom diffuses to the target walls where x-rays are produced.

are formed.

Prior to this work there were three other publications addressing the laser-induced excitation probability between the hyperfine states: the FAMU collaboration performed the calculation assuming the Fermi-golden rule with Doppler convolution while neglecting collisional effects as well as laser bandwidth [35]. This calculation was revised in Ref. [36] that corrected a mistake in the matrix element. The J-PARC collaboration calculated the laser-induced transition probability using the optical Bloch equations [37], also omitting collisional effects given their low target density [34]. Laser bandwidth was also omitted in this calculation. In contrast, decoherence effects due to collisions and laser bandwidth were considered in this work, as they reduce the transition probability by almost a factor of two at the optimal experimental conditions. Therefore, collisional rates between $\mu$p and $H_2$ were calculated in this work for the CREMA experimental conditions following the theory of Ref. [38].

The paper is organized as follows: in Sec. 2 we summarize the CREMA experimental scheme needed to understand the experimental conditions in which the laser excitation takes place. Section 3 introduces our theoretical framework based on the optical Bloch equations. We also summarize there the collisional effects and the transition matrix elements. In Sec. 4, starting from the Bloch equations we derive well-known analytical expressions valid only in certain regimes which serve to better understand the dynamics during the laser excitation and the numerical results of Sec. 5. The latter section presents the results obtained by integrating numerically the Bloch equations at various experimental conditions and discusses in detail the impact of collisions, Doppler-effects, and laser parameters.

## 2 The experimental scheme

In this section we briefly present the experimental scheme for the measurement of the HFS pursued by the CREMA collaboration, to precisely define the goal and framework of the calculation. A low-energy muon beam of about 11 MeV/c momentum is stopped in a hydrogen gas target ($\sim$ 1 mm thickness, a pressure ranging from 0.5 to 1 bar, and at a temperature ranging from 20 to 50 K), wherein a $\mu$p atom is formed in a highly excited state. The formed muonic atom quickly deexcites to the singlet ($F = 0$) state of the ground state (see Figure 1) while thermalizing within about 1 $\mu$s to the hydrogen gas temperature. When thermalized, the $\mu$p

atom is illuminated by a laser pulse at a wavelength of 6.8 $\mu$m (equivalent to a frequency of 44 THz, or 0.18 eV energy) to drive the hyperfine transitions. To enhance the laser transition probability, the laser pulse with a duration of about 20 ns and 5 mJ of energy is coupled through a 0.5 mm wide slit into a multi-pass cavity. The multiple reflections occurring in this cavity, whose average number depends mainly on the mirror reflectivity and the losses at the coupling slit, enhance the laser fluence at the position of the $\mu$p atom and increase the effective pulse length. On-resonance laser light thus excites the muonic atom from the singlet $F = 0$ to the triplet $F = 1$ sublevels. A subsequent inelastic collision with a hydrogen molecule deexcites the $\mu$p atom from the triplet back to the singlet sublevels. In this process the HFS transition energy of 0.18 eV is converted into kinetic energy: on average 0.1 eV of kinetic energy is imparted to the $\mu$p. This kinetic energy, which is much larger than the thermal energy, causes the $\mu$p atom to diffuse away from the laser-illuminated volume, reaching one of the gold-coated target walls in a time window between 100 and 300 ns after the laser excitation. When the $\mu$p atom reaches the wall, the muon is transferred to a gold atom forming muonic gold ($\mu$Au$^*$) in highly excited states. The various x-rays of MeV energy produced in the subsequent deexcitation of $\mu$Au$^*$ are used as signature of a successful laser excitation, so that the HFS resonance can be determined by counting the number of $\mu$Au x-rays as a function of the laser frequency.

In this paper, we present the calculation of the probability that a $\mu$p atom initially in the singlet state and thermalized at the temperature of the hydrogen gas will undergo the above described sequence of laser excitation and collisional deexcitation, acquiring the extra $\sim 0.1$ eV of kinetic energy needed to provide the observable signal.

## 3 Theoretical framework

### 3.1 Bloch equations

The laser excitation and the population dynamics between hyperfine states is investigated in the framework of the density-matrix formalism, using optical Bloch equations expressed as [39],

$$\frac{d\rho_{11}}{dt}(t) = -\mathrm{Im}\left(\Omega\rho_{12}e^{i\Delta t}\right) + \Gamma_{sp}\rho_{22}, \tag{1}$$

$$\frac{d\rho_{22}}{dt}(t) = \mathrm{Im}\left(\Omega\rho_{12}e^{i\Delta t}\right) - (\Gamma_i + \Gamma_{sp})\rho_{22}, \tag{2}$$

$$\frac{d\rho_{12}}{dt}(t) = \frac{i\Omega^*}{2}(\rho_{11} - \rho_{22})e^{-i\Delta t} - \frac{\Gamma_c}{2}\rho_{12}, \tag{3}$$

$$\frac{d\rho_{33}}{dt}(t) = \Gamma_i\rho_{22}, \tag{4}$$

where the detuning $\Delta = \omega_r - \omega$ is the departure of the laser angular frequency $\omega$ from the atomic resonance angular frequency $\omega_r$, $\Omega$ is the Rabi frequency, $\Gamma_i$ is the triplet deexcitation rate caused by the inelastic collisions with the H$_2$ gas leading the $\mu$p atoms to gain extra kinetic energy, $\Gamma_{sp}$ is the spontaneous radiative decay rate from the triplet state (with negligible change of the kinetic energy), and $\Gamma_c$ is the decay rate of the coherence, referred as "decoherence rate". All deexcitation and decoherence sources in equations (1)-(4) are in units of Hertz. Within our experimental conditions, $\Gamma_{sp} \ll \Gamma_i$ so that throughout this paper $\Gamma_{sp}$ can be neglected. Note that in equations (1)-(4) we also neglect the muon decay, since at our experimental conditions the muon decay rate is much smaller than the decoherence rate. The overall decay of all populations due to muon decay ($\tau_\mu = 2.2$ $\mu$s) can be treated separately simply by multiplying our results by $\exp(-t/\tau_\mu)$.

The diagonal density terms $\rho_{11}$ and $\rho_{22}$ are the populations of the thermalized singlet and (excited) triplet states, respectively, while $\rho_{12}$ is the coherence between these two states.

The third level with population $\rho_{33}$ is used to track the fraction of $\rho_{22}$ that undergoes a collisionally-induced deexcitation from the triplet to the singlet states, with consequent increase of the kinetic energy. This third level is modeled as a dark state decoupled from the laser because the additional 0.1 eV of kinetic energy (corresponding to ~1000 K), acquired by the $\mu$p in the inelastic collision, gives rise to such a large Doppler shift that the atomic resonance becomes detuned from the exciting laser frequency. Actually, within several hundred nanoseconds, the $\mu$p atoms can thermalize again through elastic $\mu$p–$H_2$ collisions. However, $\mu$p atoms with the extra kinetic energy are very likely to exit the laser-excitation volume within tens of nanoseconds. Hence, multiple cycles of laser-excitation followed by collisional-quenching are negligible at our experimental conditions, and thus modeling the third level as a dark state is appropriate.

As initial conditions for the time $t = 0$, we assume $\rho_{11} = 1$ and $\rho_{22} = \rho_{33} = 0$ since in the experiment we choose the arrival time of the laser pulse so that all $\mu$p atoms are deexcited to the singlet state and thermalized to the hydrogen gas temperature. As a consequence of $\rho_{22} = 0$, for time $t = 0$ the coherence $\rho_{12} = 0$ is also zero. In this paper, we assume that the laser excitation occurs from time $t = 0$ until $t = \tau$ at constant laser intensity $\mathcal{I}$, resulting in a laser fluence of $\mathcal{F} = \mathcal{I}\tau$ [J/cm$^2$]. The impact of the laser fluence $\mathcal{F}$ and exposure time $\tau$ on the population dynamics is evaluated throughout this paper. Exposure times of 10 ns and 100 ns are considered. Note that the actual time (and spatial) distribution of the laser intensity within the multi-pass cavity is complicated by the superpositions of the folded laser beam spreading in the cavity, however at late times it follows approximatively an exponential function with a lifetime ranging from 50 ns to 150 ns depending on the performance and geometry of the cavity.

In this work, the temporal evolution of the populations in the three levels is computed by numerically integrating the optical Bloch equations for various target conditions and laser performances. When combined with simulations of the diffusion process, muon beam and detection system, this allows optimization of the experimental setup to maximize statistical significance (signal/$\sqrt{\text{background}}$). The parameters $\Gamma_i$ and $\Gamma_c$ allow a phenomenological inclusion in the Bloch equations of the laser bandwidth and of the $\mu$p–$H_2$ elastic and inelastic collisions. The Doppler effect is accounted for in a second step by folding the results from the optical Bloch equation with a Doppler (Gaussian) profile for the (initial) thermalized $\mu$p atoms.

### 3.2 Collisional rates

The decoherence rate is given by

$$\Gamma_c = 2\pi\Delta_l + \Gamma_e^{F=0} + \Gamma_e^{F=1} + \Gamma_i + \Gamma_{\text{sp}}, \tag{5}$$

where $\Delta_l$ is the laser (FWHM) bandwidth, which contains a $2\pi$ factor to convert a bandwidth into a "rate" (see [39]). $\Gamma_i$ is the inelastic $\mu$p–$H_2$ collision rate for the excited state, $\Gamma_e^{F=1}$ is the elastic collision rate for the triplet, and $\Gamma_e^{F=0}$ is the elastic collision rate for the singlet state:

$$\Gamma_i: \quad \mu p^{F=1} + H_2 \rightarrow \mu p^{F=0} + H_2^*, \tag{6}$$

$$\Gamma_e^{F=1}: \quad \mu p^{F=1} + H_2 \rightarrow \mu p^{F=1} + H_2^*, \tag{7}$$

$$\Gamma_e^{F=0}: \quad \mu p^{F=0} + H_2 \rightarrow \mu p^{F=0} + H_2^*. \tag{8}$$

Here the $^*$ indicates possible rotational excitations of the hydrogen molecule after the collision. Note that these collisional cross-sections do not depend on the magnetic sub-states of the $\mu$p atom [40, 41].

Table 1: $\mu$p–$H_2$ collision rates in MHz for various $H_2$ target pressures ($p$) and temperatures ($T$). Two distributions of ortho- and para-$H_2$ are considered: statistical and Boltzmann.

| $p = 0.5$ bar | | | | | | |
|---|---|---|---|---|---|---|
| | $T = 22$ K | | $T = 30$ K | | $T = 50$ K | |
| | Stat. | Boltz. | Stat. | Boltz. | Stat. | Boltz. |
| $\Gamma_e^{F=0}$ | 20 | 20 | 15 | 15 | 9 | 9 |
| $\Gamma_e^{F=1}$ | 52 | 29 | 41 | 24 | 28 | 18 |
| $\Gamma_i$ | 82 | 93 | 59 | 66 | 34 | 37 |
| $p = 1$ bar | | | | | | |
| $\Gamma_e^{F=0}$ | 40 | 39 | 30 | 30 | 19 | 19 |
| $\Gamma_e^{F=1}$ | 104 | 59 | 83 | 47 | 55 | 37 |
| $\Gamma_i$ | 164 | 187 | 118 | 133 | 68 | 74 |
| $p = 2$ bar | | | | | | |
| $\Gamma_e^{F=0}$ | 79 | 79 | 61 | 61 | 38 | 37 |
| $\Gamma_e^{F=1}$ | 208 | 118 | 165 | 94 | 110 | 74 |
| $\Gamma_i$ | 328 | 374 | 235 | 265 | 137 | 148 |

All the collisional rates ($\Gamma_i$, $\Gamma_e^{F=1}$ and $\Gamma_e^{F=0}$) were obtained using

$$\Gamma = \overline{v_r \sigma(v_r)} \rho_{H_2}, \tag{9}$$

where $v_r$ is the $\mu$p–$H_2$ relative velocity and $\rho_{H_2}$ is the number density of $H_2$ molecules, which was calculated assuming that $H_2$ behaves as an ideal gas. The term $\overline{v_r \sigma(v_r)}$ represents the folding of the relative $\mu$p–$H_2$ velocity ($v_r$) distribution for thermalized $\mu$p atoms and $H_2$ molecules with the velocity-dependent cross sections $\sigma(v_r)$ for the considered scattering process [38].

These rates were calculated following Ref. [38], which uses the velocity- and spin-dependent cross sections between $\mu$p atoms and hydrogen molecules summed over the final rotational states of the $H_2$ molecules and averaged over the distribution of the initial rotational states. The spin alignment of the two nuclei of the molecule is taken into consideration when computing the rotational states and their initial populations. Two distributions for the initial rotational states were considered: statistical and Boltzmann distributions. At room temperature, the Boltzmann distribution is similar to the statistical distribution given by 75% ortho-hydrogen (odd rotational number $K$) and 25% para-hydrogen (even $K$). For low temperatures and at equilibrium, the Boltzmann distribution differs considerably from the statistical distribution, e.g. at 22 K, basically all molecules have $K = 0$. However, if the gas is quickly cooled down, the ratio ortho- to para-hydrogen existing at high temperature is retained so that 75% of the molecules have $K = 1$ and 25% have $K = 0$. The Boltzmann distribution is only reached after a long time because the conversion from $K = 1$ to $K = 0$ rotational states is a very slow process having a rate of about 2% per week at the normal temperature and pressure [42]. The actual rotational distribution in the experimental conditions is therefore bounded between these two extreme cases. For this reason, we give in Table 1 values of the collision rates for statistical and Boltzmann distributions.

Table 2: Values of $\mathcal{M}$ and $\Omega/\sqrt{\mathcal{I}}$ for Lamb shift and HFS transitions in $\mu$p and $\mu^3$He$^+$. The analytical expressions for the $2s-2p$ matrix elements agree with [43]. $a_\mu$ is the muonic Bohr radius.

| Atom | Transition | $\mathcal{M}$ [m] | $\frac{\Omega}{\sqrt{\mathcal{I}}}$ [m/$\sqrt{\text{Js}}$] |
|---|---|---|---|
| $\mu$p | $2s^{F=1} \to 2p_{3/2}^{F=2}$ | $\sqrt{5}a_\mu = 6.367 \times 10^{-13}$ | $2.65 \times 10^4$ |
| $\mu^3$He$^+$ | $2s^{F=1} \to 2p_{3/2}^{F=2}$ | $\frac{\sqrt{5}}{2}a_\mu = 2.969 \times 10^{-13}$ | $1.24 \times 10^4$ |
| $\mu$p | $1s^{F=0} \to 1s^{F=1}$ | $\frac{\hbar}{4m_\mu c}\left(g_\mu + \frac{m_\mu}{m_p}g_p\right)$ $= 1.228 \times 10^{-15}$ | $5.12 \times 10^1$ $^a$ |
| $\mu^3$He$^+$ | $1s^{F=1} \to 1s^{F=0}$ | $\frac{\hbar}{4\sqrt{3}m_\mu c}\left(g_\mu + \frac{m_\mu}{m_{\text{He}}}g_{\text{He}}\right)$ $= 4.965 \times 10^{-16}$ | $2.07 \times 10^1$ |

$^a$ $1.77 \times 10^1$ m/$\sqrt{\text{Js}}$ according to Ref. [37]

## 3.3 Rabi frequencies and matrix elements

The Rabi frequency $\Omega = \sqrt{\frac{8\pi\alpha\mathcal{I}}{\hbar}}\mathcal{M}$ included in the Bloch equations quantifies the coupling strength between the laser with intensity $\mathcal{I}$, and the atomic transition. Here, $\alpha$ is the fine structure constant and $\hbar$ the reduced Planck constant. We evaluated the matrix element $\mathcal{M}$ for the M1 HFS transitions of $\mu$p and $\mu^3$He$^+$ in the ground state. To verify the calculations, we also evaluated the matrix elements for the already measured $2s-2p$ transitions in $\mu$p and $\mu^3$He$^+$, where literature values of saturation fluences are available [43].

Because the magnetic substates of the initial state can be assumed to be statistically populated, the matrix element is averaged over the initial magnetic sub-states $m$:

$$\mathcal{M}^2 = \frac{1}{2F_i + 1}\sum_{m,m'}\left|\mathcal{M}^{(m,m')}\right|^2 . \tag{10}$$

It is also summed over all possible final magnetic sub-states $m'$ since the final magnetic sub-states are not detected (resolved) in our experimental scheme. The matrix elements $\mathcal{M}^{(m,m')}$ for the $2s-2p$ (E1-type) and $1s$-HFS (M1-type) transitions are given by [35, 44]

$$\mathcal{M}_{\text{E1}}^{(m,m')} = \left\langle 2p\, F'\, m'\, |\boldsymbol{r}\cdot\hat{\boldsymbol{\varepsilon}}|\, 2s\, F\, m\right\rangle , \tag{11}$$

$$\mathcal{M}_{\text{M1}}^{(0,m')} = \frac{1}{2m_\mu c}\left\langle F'=1\, m'\, \left|(g_\mu \boldsymbol{S} + g_p\frac{m_\mu}{m_p}\boldsymbol{I})\cdot\hat{\boldsymbol{k}}\times\hat{\boldsymbol{\varepsilon}}\right|\, F=0\, 0\right\rangle , \tag{12}$$

where $m_\mu$ is the muon mass, $m_p$ the proton mass, $\hat{\boldsymbol{k}}$ the laser wavevector, $\hat{\boldsymbol{\varepsilon}}$ the laser polarization, $\boldsymbol{S}$ the spin operator of the muon, $\boldsymbol{I}$ the spin operator of the proton, and $g_\mu$ and $g_p$ the $g$-factor of the muon ($\simeq 2.00$) and proton ($\simeq 5.58$), respectively. See Appendix A for more details about the evaluation of these matrix elements.

Table 2 summarizes the numerical values of the (analytical) matrix elements $\mathcal{M}$ and $\Omega/\sqrt{\mathcal{I}}$ for the HFS and the most intense Lamb shift transitions for $\mu$p and $\mu^3$He$^+$. While the matrix elements for the $2s-2p$ transitions are in agreement with published values [43], the $\Omega/\sqrt{\mathcal{I}}$ value obtained for the $\mu$p HFS of $5.12 \times 10^1$ m/$\sqrt{\text{Js}}$ is in disagreement with the value of $1.77 \times 10^1$ m/$\sqrt{\text{Js}}$ published in Ref. [37]. The difference of $\sqrt{\frac{3}{2}}\left(\frac{g_\mu m_p + g_p m_\mu}{m_p + m_\mu}\right) \approx 2.9$ is traced back to a miscalculation of the matrix elements in Ref. [37] (see Appendix A), and by not considering the proton spin-flip.

Table 3: Saturation fluences for laser bandwidths of $\Delta_l = 10$ MHz ($\mathcal{F}_{\text{sat}}^{\Delta_l \to 10}$) and $\Delta_l = 100$ MHz ($\mathcal{F}_{\text{sat}}^{\Delta_l \to 100}$) for three transitions in muonic atoms, accounting for collisional effects. For comparison we also give the saturation fluence neglecting decoherence effects ($\mathcal{F}_{\text{sat}}^{\Gamma_c \to 0}$). The resonance frequencies $\nu_r = \omega_r/2\pi$ for the three transitions have been taken from Refs. [5, 45, 46]. The spontaneous decay rates $\Gamma_{\text{sp}}$ of the $2p$ states were taken from [47] and [48], while for the ground state triplet in $\mu$p it is calculated here. The decoherence rates $\Gamma_c$ have been obtained using equation (5) with the values from Table 1 assuming a Boltzmann distribution between ortho- and para-H$_2$ (for a statistical distribution see Appendix B). $\sigma_D$ is the Doppler standard deviation (15).

| Atom | Transition | $T$ [K] | $p$ [bar] | $\nu_r$ [THz] | $\Gamma_{\text{sp}}$ [MHz] | $\Gamma_c^{\Delta_l \to 10}$ [MHz] | $\Gamma_c^{\Delta_l \to 100}$ [MHz] | $\sigma_D$ [MHz] | $\mathcal{F}_{\text{sat}}^{\Gamma_c \to 0}$ [J/cm$^2$] | $\mathcal{F}_{\text{sat}}^{\Delta_l \to 10}$ [J/cm$^2$] | $\mathcal{F}_{\text{sat}}^{\Delta_l \to 100}$ [J/cm$^2$] |
|---|---|---|---|---|---|---|---|---|---|---|---|
| $\mu$p | $2s^{F=1} \to 2p_{3/2}^{F=2}$ | 300 | 0.001 | 49.9 | $1.16 \times 10^5$ | | $1.16 \times 10^5$ | $2.48 \times 10^2$ | | | 0.0165 |
| $\mu^3$He$^+$ | $2s^{F=1} \to 2p_{3/2}^{F=2}$ | 300 | 0.004 | 379 | $2.00 \times 10^6$ | | $2.00 \times 10^6$ | $1.13 \times 10^3$ | | | 1.304 |
| $\mu$p | $1s^{F=0} \to 1s^{F=1}$ | | | 44.2 | $1.23 \times 10^{-11}$ | | | | | | |
| | | 22 | 0.5 | | | | 205 | 770 | 60 | 23 | 28 | 44 |
| | | 22 | 1 | | | | 348 | 913 | 60 | 23 | 32 | 49 |
| | | 22 | 2 | | | | 633 | 1198 | 60 | 23 | 40 | 58 |
| | | 30 | 0.5 | | | | 168 | 733 | 70 | 27 | 31 | 47 |
| | | 30 | 1 | | | | 273 | 838 | 70 | 27 | 34 | 50 |
| | | 50 | 0.5 | | | | 128 | 693 | 90 | 35 | 37 | 53 |
| | | 50 | 1 | | | | 192 | 757 | 90 | 35 | 39 | 55 |

## 3.4 Doppler broadening

While the collisional effects and the laser bandwidth are already accounted for in the Bloch equations, and eventually give rise to a Lorentzian profile (see Sec. 4.1), the Doppler broadening needs to be treated separately.

We include the Doppler effect by convoluting the population $\rho_{33}(\omega)$ as obtained from the Bloch equations with a Gaussian distribution describing the Doppler profile:

$$\bar{\rho}_{33}(\omega) = \int_{-\infty}^{\infty} \rho_{33}(\omega') \frac{1}{\sqrt{2\pi}\gamma_D} \exp\left(-\frac{(\omega-\omega')^2}{2\gamma_D^2}\right) d\omega', \tag{13}$$

with $\gamma_D$ being given by

$$\gamma_D = \omega_r \sqrt{\frac{kT}{(m_\mu + m_p)c^2}} \quad \simeq \quad 7.98 \times 10^7 \sqrt{T} \text{ [rad/s]}, \tag{14}$$

$$\sigma_D = \frac{\gamma_D}{2\pi} \quad \simeq \quad 12.7\sqrt{T} \text{ [MHz]}, \tag{15}$$

so that the Doppler standard deviation (MHz) is $\sigma_D$. $k$ is the Boltzmann constant, $c$ the speed of light, and $T$ the temperature in Kelvin of the thermalized $\mu$p atom in the singlet state.

## 4 Analytical expressions for two limiting regimes

Before integrating numerically the three-level Bloch equations and investigating how collisional and Doppler effects impact the $\rho_{33}$ population, it is interesting to derive analytical expressions from our formalism to reproduce well-known results valid for low and high Rabi frequencies, where low and high are relative to the other frequencies involved in the problem.

### 4.1 Fermi-golden-rule regime

The laser-induced rate of population transfer $R(t, \omega)$ from $\rho_{11}$ to $\rho_{22}$ is given by the first term of the second Bloch equation (2),

$$R(t, \omega) = \text{Im}\left(\Omega \rho_{12} e^{i \Delta t}\right). \tag{16}$$

In the limit of low transition probability ($\Omega \ll \Gamma_c$), and sufficiently long time ($t \gg 1/\Gamma_c$) to obtain stable $\rho_{22}$ population, this rate converts to the Lorentzian form of the Fermi-golden rule [39],

$$R_{\text{F}}^{(\text{L})}(\omega) = \frac{|\Omega|^2}{4} \frac{\Gamma_c}{\left(\Delta^2 + \left(\frac{\Gamma_c}{2}\right)^2 + \frac{|\Omega|^2 \Gamma_c}{2 \Gamma_i}\right)}. \tag{17}$$

This result is obtained using the analytical solutions of $\rho_{22}$ and $\rho_{12}$ of Refs. [49–51]. Note that the Fermi-golden-rule approach does not contain population dynamics; it is a static approach.

The Doppler effect can be included by performing a convolution similar to equation (13)

$$\bar{R}_{\text{F}}^{(\text{L})}(\omega) = \int_{-\infty}^{\infty} R_{\text{F}}^{(\text{L})}(\omega') \frac{1}{\sqrt{2\pi}\gamma_D} \exp\left(-\frac{(\omega - \omega')^2}{2\gamma_D^2}\right) d\omega', \tag{18}$$

resulting in a combined laser-excitation and collisional deexcitation probability of

$$\bar{\rho}_{33}(\omega) \approx \hat{\rho}_{33}(\omega) = \int_0^\tau \bar{R}_{\text{F}}^{(\text{L})}(\omega) dt = \frac{\mathcal{F}}{\mathcal{F}_{\text{sat}}(\omega)}, \tag{19}$$

where $\mathcal{F}_{\text{sat}}$ is the saturation fluence and $\bar{\rho}_{33}(\omega)$ is the obtained Voigt profile that accounts for Lorentzian (homogeneous) and Gaussian (inhomogeneous) broadenings. The hat-symbol on the third level population $\hat{\rho}_{33}$ is used to denote that it was computed in the Fermi-golden-rule approximation and including Doppler effects.

Since the Voigt function at resonance has an analytical solution, at resonance we can find an analytical expression for $\mathcal{F}_{\text{sat}}(\omega_r)$:

$$\mathcal{F}_{\text{sat}}(\omega_r) = \frac{\hbar \gamma_D^2}{\pi^{3/2} \alpha \Gamma_c \mathcal{M}^2} \frac{\varpi e^{-\varpi^2}}{\text{Erfc}(\varpi)}, \tag{20}$$

where $\varpi = \frac{\sqrt{2}\Gamma_c}{4\gamma_D}$ and Erfc is the complementary error function. For $\Gamma_c \ll \sigma_D$ the saturation fluence becomes

$$\mathcal{F}_{\text{sat}}^{\Gamma_c \to 0}(\omega_r) = \frac{\hbar \sigma_D}{(2\pi)^{3/2} \alpha \mathcal{M}^2} \approx \frac{\sqrt{T}}{0.2} \; [\text{J/cm}^2]. \tag{21}$$

In this limit, the laser transition probability becomes

$$\hat{\rho}_{33}^{\Gamma_c \to 0} \approx \frac{0.2}{\sqrt{T}} \mathcal{F}, \tag{22}$$

where $T$ and $\mathcal{F}$ are values of temperature and fluence in units of K and J/cm$^2$, respectively. This result is in agreement with the excitation probability given in Ref. [36].

Table 3 summarizes the saturation fluences $\mathcal{F}_{\text{sat}}$ calculated from equation (20), for Lamb shift transitions in $\mu$p and $\mu$He$^+$, as well as for the HFS transition in $\mu$p for possible experimental conditions. Obtained values of $\mathcal{F}_{\text{sat}}$ for Lamb shift transitions are in agreement with Ref. [43]. Four orders of magnitude separate the $\mathcal{F}_{\text{sat}}$ for the Lamb shift and the HFS transitions in $\mu$p, emphasizing the laser technology leap needed to accomplish the hyperfine experiment. Values of $\mathcal{F}_{\text{sat}}^{\Gamma_c \to 0}$ given by equation (21) are also shown together with $\mathcal{F}_{\text{sat}}^{\Delta_l \to 10}$ and

$\mathcal{F}_{\text{sat}}^{\Delta_l \to 100}$, which are the saturation fluences for $\Delta_l = 10$ MHz and 100 MHz, respectively. The comparison of $\mathcal{F}_{\text{sat}}^{\Delta_l \to 10}$ to $\mathcal{F}_{\text{sat}}^{\Delta_l \to 100}$ highlights the impact of the laser bandwidth, while comparison of $\mathcal{F}_{\text{sat}}^{\Delta_l \to 10}$ to $\mathcal{F}_{\text{sat}}^{\Gamma_c \to 0}$ allows to appreciate the impact of the collisional effects.

Table 5 in Appendix B shows that within our experimental conditions, the decoherence rates and the saturation fluences are only marginally affected by the assumed ortho- to para-$H_2$ distributions. Therefore, throughout this study, we assumed a Boltzmann distribution.

## 4.2 Rabi-oscillation regime: Two-level Bloch equations

For $\Gamma_c < \Omega$, the approximation of equation (19) is no longer valid. In this limit, the $\rho_{22}$ population saturates and oscillates according to

$$\rho_{22} = \frac{|\Omega|^2}{|\Omega|^2 + \Delta^2} \sin^2\left(\frac{\tau}{2}\sqrt{|\Omega|^2 + \Delta^2}\right), \tag{23}$$

which on resonance simplifies to

$$\rho_{22} = \sin^2\left(\sqrt{\frac{8\pi\alpha\mathcal{F}\tau}{\hbar}}\mathcal{M}\right). \tag{24}$$

These are the well-known analytical solutions of the two-level Bloch equations valid in the Rabi-oscillation regime when broadening sources can be neglected [39], i.e., when the Rabi frequency dominates over the broadening rates $\Omega \gg \Gamma_c$ and $\Omega \gg \sigma_D$.

Reference [37] calculates the transition probability between hyperfine states in $\mu$p using equation (23), hence without decoherence (collisional or laser bandwidth) effects. Doppler broadening was included in a second step in their calculations as a not-well-specified numerical average.

# 5 Results and discussion

## 5.1 $\bar{\rho}_{33}$ at the experimental conditions

Simulating the time-evolution of state populations using Bloch equations has improved laser spectroscopy of molecules [52], muonium [53] and highly charged ions [54]. Similarly, the findings of population dynamics obtained from Eq. (1)-(4) can be used to optimize the target conditions ($T$ and $p$) of the HFS experiment.

We present here the $\bar{\rho}_{33}$ populations for laser parameters and hydrogen target conditions in the current region of interest for the CREMA HFS experiment. These populations obtained from the numerical integration of the Bloch equations are shown by the dashed curves in Figure 3 and compared with the approximations of equation (19), shown as solid lines. Already for fluences larger than 5 J/cm$^2$ the full-numerical calculations need to be used for accurate predictions. At the prospected optimal conditions of $T = 22$ K and $p = 0.5$ bar (based on preliminary simulations of the diffusion of the $\mu$p atom in the $H_2$ gas target), for $\Delta_l = 10$ MHz, the analytical prediction overestimate the transition probability by about 30% and 46% for fluences of 10 J/cm$^2$ and 20 J/cm$^2$, respectively. For $\Delta_l = 100$ MHz, this difference is reduced by 17% and 26% for the same fluences, respectively. Note that fluences up to 20 J/cm$^2$ are achievable locally in our multi-pass cavity. The importance of the dynamics is also highlighted by the small but non-negligible dependence of $\bar{\rho}_{33}$ on the exposure time $\tau$.

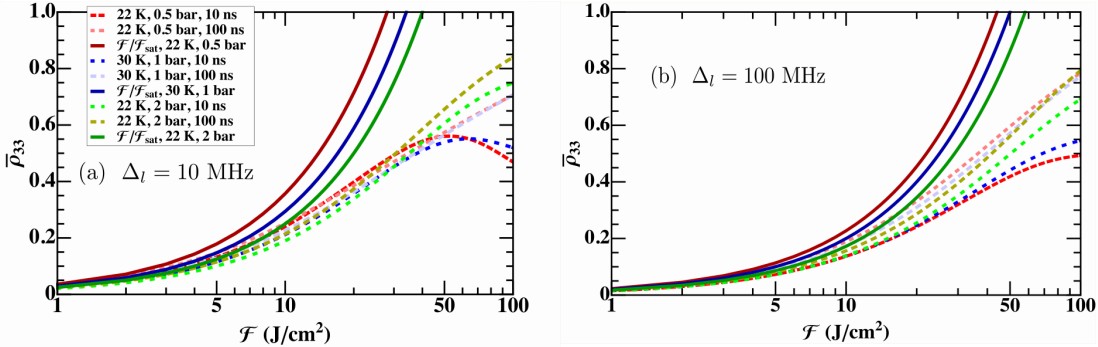

Figure 3: (Color online) $\bar{\rho}_{33}$ population versus laser fluence $\mathcal{F}$ for various $H_2$ target conditions, two laser bandwidths ($\Delta_l = 10$ MHz and $\Delta_l = 100$ MHz), and two exposure times ($\tau = 10$ ns and $\tau = 100$ ns) in the range of interest for the CREMA HFS experiment. We use the collision rates of Table 1 assuming Boltzmann distribution between ortho- and para-$H_2$. The solid lines represent the results from the analytical expression of equation (19) that neglects population dynamics. The saturation fluences $\mathcal{F}_{\text{sat}}$ are taken from Table 3.

## 5.2 Resonance profile at the experimental conditions

The HFS resonance has to be searched for in a region of about 40 GHz ($\pm 3\sigma$) given by the uncertainty of the theoretical predictions arising mainly from the uncertainty of the two-photon exchange contribution [13–23]. The resonance is scanned by counting the number of x-rays for a few hours at a given laser frequency. Then the laser frequency is shifted by a fixed amount and the x-ray counting is performed anew. This is repeated until a statistically significant deviation of the number of x-ray counts from the background level has been observed. Given the large search range, the long time (few hours) per frequency point necessary to observe a statistically significant deviation above background, and the limited access at the PSI accelerator, it is important to optimize the frequency step used to search for the resonance. For this reason, the linewidth of the targeted resonance has to be known precisely.

Figure 4 shows some line profiles obtained from numerical integration at conditions relevant for the CREMA experiment. The figure clearly shows that the reduction of the on-resonance excitation probability due to collisional effects is correlated with a broadening of the line-shape.

For fluences $\mathcal{F} \lesssim 20$ J/cm$^2$, the Rabi frequency is sufficiently small so that this power broadening can be neglected. In this regime the line shape is thus a Lorentzian profile with a FWHM of approximately [39]

$$\Gamma_L^2 = \Gamma_c^2 + 2|\Omega|^2 \frac{\Gamma_c}{\Gamma_i} \,. \tag{25}$$

When including the Doppler effect, a Voigt profile is obtained with a FWHM-linewidth $\hat{\Gamma}_V$ given approximately by [55]:

$$\hat{\Gamma}_V = 0.53\Gamma_L + \sqrt{0.22\Gamma_L^2 + 5.54\gamma_D^2} \,. \tag{26}$$

The $\hat{\Gamma}_V$, calculated using the analytical expressions of Eqs. (25) and (26) are compared in Table 4 to the FWHM linewidth obtained by numerically integrating the optical Bloch equations. For completeness, Table 4 also summarizes the $\hat{\rho}_{33}(\omega_r)$ and $\bar{\rho}_{33}(\omega_r)$ as obtained from the analytical expression of equation (19) and from the full numerical calculation, respectively. Here, a Boltzmann distribution between ortho- and para-hydrogen was assumed while the respective values for a statistical distribution are listed in Appendix B. Their comparison shows a negligible dependence on the type of distribution assumed. Tables 4, 6 and 7 provide the

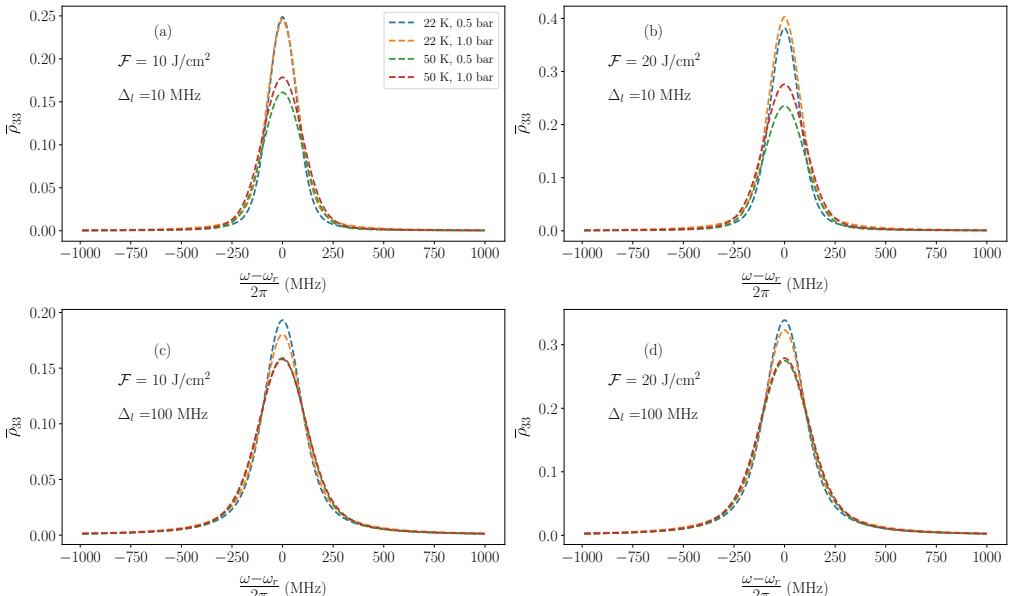

Figure 4: (Color online) $\bar{\rho}_{33}$ versus the laser angular frequency $\omega$ (line-shape) calculated by numerically integrating the Bloch equation for $\tau = 100$ ns, at various target conditions assuming Boltzmann distributions between ortho- and para-$H_2$: (a) $\mathcal{F} = 10$ J/cm$^2$ and $\Delta_l = 10$ MHz; (b) $\mathcal{F} = 20$ J/cm$^2$ and $\Delta_l = 10$ MHz; (c) $\mathcal{F} = 10$ J/cm$^2$ and $\Delta_l = 100$ MHz; (d) $\mathcal{F} = 20$ J/cm$^2$ and $\Delta_l = 100$ MHz. Values of $\bar{\rho}_{33}$ on resonance and FWHM linewidth can be found in Table 4.

complete information needed to quantify the combined probability of laser-excitation followed by collisional deexcitation for the HFS experiment of the CREMA collaboration.

## 6 Conclusion

In view of the upcoming HFS experiment of the CREMA collaboration, we calculated the combined probability that a thermalized $\mu$p atom undergoes laser excitation from the singlet to the triplet states followed by a collisional deexcitation, where it acquires on average about 0.1 eV extra kinetic energy. This calculation was performed accounting for collisional and Doppler effects. The collisional effects together with the laser properties such as bandwidth, exposure time, and fluence have been accounted for directly in the three-level Bloch equations, while the Doppler broadening has been accounted for in a second step by convoluting the results from the Bloch equations with the Doppler profile.

We derived simple analytical expressions valid in limiting regimes (Fermi-golden rule and Rabi-oscillations). Given the small Rabi frequency relative to the homogeneous broadening, the proposed HFS experiment lies more closely to the Fermi-golden-rule regime.

The collisional effects play an important role: while the inelastic collisions trigger a successful laser transition, the elastic collisions decrease the laser excitation probability to the triplet state. In the experimental conditions of the CREMA HFS experiment, (around 0.5 bar, 22 K, $\Delta_l = 100$ MHz) the decoherence effects decrease the transition probability for small flueces by about a factor of two (see Table 3).

For fluences reached in the optical cavity (up to 20 J/cm$^2$ locally), the dynamical effects beyond the Fermi approximation captured by the numerical integration of the Bloch equations

Table 4: $\hat{\Gamma}_V$ from the analytical expression of equation (26), FWHM line-width $\Gamma_V$ extracted from fitting the line-profile obtained from numerical integration of the Bloch equations for various $\omega$, $\hat{\rho}_{33}(\omega_r)$ from the analytical expression of equation (19) and $\bar{\rho}_{33}(\omega_r)$ from the numerical integration of the Bloch equations. $\tau = 100$ ns and Boltzmann distribution between ortho- and para-H$_2$ are assumed (for $\tau = 10$ ns and a statistical distribution see Appendix B). The symbol "-" indicates that $\hat{\rho}_{33} > 1$, which is non-physical.

| $p$ [bar] | $T$ [K] | $\Delta_l$ [MHz] | $\Gamma_c$ [MHz] | $\sigma_D$ [MHz] | $\mathcal{F} = 10$ J/cm$^2$ ($\Omega = 8$ MHz) | | | | $\mathcal{F} = 20$ J/cm$^2$ ($\Omega = 11$ MHz) | | | | $\mathcal{F} = 50$ J/cm$^2$ ($\Omega = 18$ MHz) | | | |
|---|---|---|---|---|---|---|---|---|---|---|---|---|---|---|---|---|
| | | | | | $\hat{\Gamma}_V$ [MHz] | $\Gamma_V$ [MHz] | $\hat{\rho}_{33}$ | $\bar{\rho}_{33}$ | $\hat{\Gamma}_V$ [MHz] | $\Gamma_V$ [MHz] | $\hat{\rho}_{33}$ | $\bar{\rho}_{33}$ | $\hat{\Gamma}_V$ [MHz] | $\Gamma_V$ [MHz] | $\hat{\rho}_{33}$ | $\bar{\rho}_{33}$ |
| 0.5 | 22 | 100 | 770 | 60 | 220 | 228 | 0.23 | 0.19 | 223 | 232 | 0.45 | 0.33 | 231 | 267 | - | 0.59 |
| 1 | 22 | 100 | 913 | 60 | 235 | 242 | 0.21 | 0.18 | 236 | 256 | 0.41 | 0.32 | 241 | 283 | - | 0.60 |
| 0.5 | 22 | 10 | 205 | 60 | 161 | 167 | 0.36 | 0.25 | 163 | 176 | 0.71 | 0.38 | 169 | 190 | - | 0.57 |
| 1 | 22 | 10 | 348 | 60 | 174 | 183 | 0.31 | 0.25 | 175 | 187 | 0.63 | 0.40 | 178 | 208 | - | 0.64 |
| 0.5 | 30 | 100 | 733 | 70 | 239 | 247 | 0.21 | 0.17 | 243 | 251 | 0.43 | 0.30 | 254 | 277 | - | 0.54 |
| 1 | 30 | 100 | 838 | 70 | 249 | 256 | 0.20 | 0.17 | 251 | 269 | 0.40 | 0.31 | 257 | 287 | - | 0.57 |
| 0.5 | 30 | 10 | 168 | 70 | 182 | 189 | 0.32 | 0.21 | 185 | 194 | 0.65 | 0.32 | 191 | 208 | - | 0.48 |
| 1 | 30 | 10 | 273 | 70 | 190 | 199 | 0.30 | 0.22 | 192 | 204 | 0.60 | 0.36 | 196 | 226 | - | 0.56 |
| 0.5 | 50 | 100 | 693 | 90 | 283 | 288 | 0.19 | 0.15 | 290 | 295 | 0.38 | 0.28 | 308 | 316 | 0.95 | 0.42 |
| 1 | 50 | 100 | 757 | 90 | 287 | 294 | 0.18 | 0.15 | 290 | 299 | 0.37 | 0.27 | 300 | 328 | 0.91 | 0.49 |
| 0.5 | 50 | 10 | 127 | 90 | 228 | 230 | 0.27 | 0.16 | 233 | 236 | 0.53 | 0.23 | 240 | 249 | - | 0.35 |
| 1 | 50 | 10 | 192 | 90 | 231 | 236 | 0.26 | 0.18 | 231 | 240 | 0.51 | 0.27 | 240 | 256 | - | 0.43 |

are significant, as can be seen by comparing the solid and dashed lines in Figure 3 and $\hat{\rho}_{33}$ and $\bar{\rho}_{33}$ in Table 4. We also demonstrate that $\bar{\rho}_{33}$ is not significantly affected by the assumed distribution of the hydrogen rotational states.

For the a fluence of 10 J/cm$^2$, 100 MHz laser bandwidth and 100 ns laser exposure, 19% of the $\mu$p atoms exposed to the laser light acquires the extra 0.1 eV kinetic energy for a hydrogen gas target at 22 K temperature and 0.5 bar pressure.

This paper is the first in a set of studies dedicated to the $\mu$p HFS experiment of the CREMA collaboration. In conjunction with studies of the $\mu$p diffusion in the target gas, the optical multi-pass cavity and the detection system which will be published elsewhere, this allows to optimize the hydrogen target and define the specifications for the laser system.

# Acknowledgements

We acknowledge the support of the following grants: Fundação para a Ciência e a Tecnologia (FCT), Portugal, and FEDER through COMPETE in the framework of project numbers PTDC/FIS-AQM/29611/2017 and UID/04559/2020 (LIBPhys), and contracts No. SFRH/BPD/92329/2013 and No. PD/BD/128324/2017; Deutsche Forschungsgemeinschaft (DFG, German Research Foundation) under Germany's Excellence Initiative EXC 1098 PRISMA (194673446), Excellence Strategy EXC PRISMA+ (390831469) and DFG/ANR Project LASIMUS (DFG Grant Agreement 407008443); The French National Research Agency with project ANR-18-CE92-0030-02; The European Research Council (ERC) through CoG. #725039, and the Swiss National Science Foundation through the projects SNF 200021_165854 and SNF 200020_197052.

# A   Hyperfine dipole matrix elements

Evaluation of the matrix elements follows similar methods as done in recent works [48,56–58]. The hyperfine matrix elements (11) and (12) are evaluated with standard angular reduction methods, which start by expanding the electric dipole $r \cdot \hat{\varepsilon}$, and the magnetic dipole $S \cdot \hat{k} \times \hat{\varepsilon}$ operators (similarly with $I \cdot \hat{k} \times \hat{\varepsilon}$) in a spherical basis. After considering the overall atomic state being the product coupling of the nucleus and electron angular momenta, and using the Wigner-Eckart theorem, the matrix element for the irreducible rank-one tensor $T_\lambda$ (components $\lambda = \pm 1$) of the spherical expansion is given by [44,59]

$$
\begin{aligned}
\langle \beta' F' m' J' | T_\lambda | \beta F m J \rangle &= (-1)^{F'+I+F+1+J'-m'} \sqrt{[F,F']} \begin{pmatrix} F' & 1 & F \\ -m' & \lambda & m \end{pmatrix} \\
&\times \begin{Bmatrix} J & I & F \\ F' & 1 & J' \end{Bmatrix} \langle \beta' J' \| T \| \beta J \rangle .
\end{aligned}
\tag{27}
$$

Here, $T_\lambda = r_\lambda$ and $T_\lambda = S_\lambda$ are for the $2s - 2p$ and HFS cases, respectively. Both final and initial states are defined by $|f\rangle = \left| \beta' F' m' J' \right\rangle$ and $|i\rangle = |\beta F m J\rangle$, where $F$ is the total angular momentum , $m$ is its projection, and $J$ is the total angular momentum of $\mu$. $\beta$ represents further quantum numbers, such as $\ell$ and $n$. The notation $[j_1, j_2, ...]$ stands for $(2j_1 + 1)(2j_2 + 1)...$ The reduced matrix elements for E1 and M1 are given by

$$
\langle 2p J' \| r \| 2s, J = 1/2 \rangle = \sqrt{[1/2, J']} \begin{pmatrix} J' & 1 & 1/2 \\ 1/2 & 0 & -1/2 \end{pmatrix} \int P'_{2p} P_{2s} \, r \, dr ,
\tag{28}
$$

and

$$
\langle \beta' J' | S | \beta J \rangle = \frac{\sqrt{6}}{2} \delta(\ell, \ell') (-1)^{J'-1/2} \sqrt{[J, J']} \begin{Bmatrix} 1/2 & \ell & J \\ J' & 1 & 1/2 \end{Bmatrix} \hbar ,
\tag{29}
$$

for the $r$ and $S$ operators, respectively. The functions $P'_{2p}$ and $P_{2s}$ in equation (28) are hydrogenic radial wavefunctions.

After performing the averaging over initial magnetic substates $m$ and summation over final substates $m'$, the matrix element are given by

$$
\mathcal{M}_{\text{E1}} = \frac{1}{\sqrt{3}} \sqrt{[F', J, J']} \begin{Bmatrix} J & I & F \\ F' & 1 & J' \end{Bmatrix} \begin{pmatrix} J' & 1 & J \\ 1/2 & 0 & -1/2 \end{pmatrix} \int P'_{2p} P_{2s} \, r \, dr ,
\tag{30}
$$

$$
\mathcal{M}_{\text{M1}} = \frac{1}{\sqrt{8}} \sqrt{[F', F]} \begin{Bmatrix} 1/2 & I & F \\ F' & 1 & 1/2 \end{Bmatrix} \left( \frac{g_\mu \hbar}{m_\mu c} + \frac{g_p \hbar}{m_p c} \right) ,
\tag{31}
$$

for the Lamb shift and HFS, respectively. While evaluating the matrix elements, the overall contribution of light polarization is reduced to $|\hat{\varepsilon}|^2 = 1$, thus suppressing any dependence on the laser polarization (circular or linear) used. This is expected since the initial state is not polarized, and the final state polarization is not observed. Equation (31) is equivalent to equation (2) of Ref. [37] with $g_\mu = 2$, $g_p = 0$ and $1/c \rightarrow e$ (different units), however we obtain a value of $e\hbar/(2mc)$ instead of $e\hbar/(2\sqrt{6}mc)$ after its evaluation.

# B   Results for a statistical distribution and $\tau = 10$ ns

Table 5 compares $\mathcal{F}_{\text{sat}}$ and $\Gamma_c$ for statistical and Boltzmann distributions between ortho- and para-hydrogen, while Table 6 lists the transition probabilities $\bar{\rho}_{33}(\omega_r)$ and linewidths, similar

Table 5: Decoherence rates and saturation fluences for statistical and Boltzmann distributions between ortho- and para-hydrogen, and for $\Delta_l = 100$ MHz.

| $T$ [K] | $p$ [bar] | $\Gamma_c^{\text{Boltz.}}$ [MHz] | $\Gamma_c^{\text{Stat.}}$ [MHz] | $\mathcal{F}_{\text{sat}}^{\text{Boltz.}}$ [J/cm$^2$] | $\mathcal{F}_{\text{sat}}^{\text{Stat.}}$ [J/cm$^2$] |
|---|---|---|---|---|---|
| 22 | 0.5 | 771 | 782 | 44 | 45 |
| 22 | 1 | 913 | 936 | 49 | 49 |
| 22 | 2 | 1198 | 1244 | 58 | 59 |
| 30 | 0.5 | 733 | 743 | 47 | 47 |
| 30 | 1 | 838 | 859 | 50 | 50 |
| 50 | 0.5 | 693 | 699 | 53 | 53 |
| 50 | 1 | 757 | 770 | 55 | 55 |

Table 6: Same as Table 4 but for a statistical ortho- to para-distribution of the initial rotational states of $H_2$ molecules.

| $p$ [bar] | $T$ [K] | $\Delta_l$ [MHz] | $\Gamma_c$ [MHz] | $\sigma_D$ [MHz] | $\mathcal{F} = 10$ J/cm$^2$ ($\Omega = 8$ MHz) $\hat{\Gamma}_V$ [MHz] | $\Gamma_V$ [MHz] | $\hat{\rho}_{33}$ | $\bar{\rho}_{33}$ | $\mathcal{F} = 20$ J/cm$^2$ ($\Omega = 11$ MHz) $\hat{\Gamma}_V$ [MHz] | $\Gamma_V$ [MHz] | $\hat{\rho}_{33}$ | $\bar{\rho}_{33}$ | $\mathcal{F} = 50$ J/cm$^2$ ($\Omega = 18$ MHz) $\hat{\Gamma}_V$ [MHz] | $\Gamma_V$ [MHz] | $\hat{\rho}_{33}$ | $\bar{\rho}_{33}$ |
|---|---|---|---|---|---|---|---|---|---|---|---|---|---|---|---|---|
| 0.5 | 22 | 100 | 782 | 60 | 222 | 231 | 0.22 | 0.18 | 225 | 234 | 0.45 | 0.32 | 235 | 269 | - | 0.58 |
| 1 | 22 | 100 | 936 | 60 | 238 | 246 | 0.20 | 0.18 | 240 | 259 | 0.40 | 0.32 | 245 | 287 | - | 0.59 |
| 0.5 | 22 | 10 | 217 | 60 | 162 | 169 | 0.35 | 0.25 | 165 | 177 | 0.71 | 0.38 | 171 | 193 | - | 0.57 |
| 1 | 22 | 10 | 371 | 60 | 176 | 182 | 0.31 | 0.24 | 177 | 191 | 0.62 | 0.40 | 182 | 213 | - | 0.64 |
| 0.5 | 30 | 100 | 744 | 70 | 241 | 249 | 0.21 | 0.17 | 245 | 253 | 0.43 | 0.29 | 258 | 279 | - | 0.53 |
| 1 | 30 | 100 | 859 | 70 | 251 | 259 | 0.20 | 0.17 | 254 | 270 | 0.40 | 0.30 | 261 | 290 | 0.99 | 0.56 |
| 0.5 | 30 | 10 | 178 | 70 | 183 | 190 | 0.32 | 0.21 | 186 | 196 | 0.64 | 0.32 | 194 | 211 | - | 0.48 |
| 1 | 30 | 10 | 293 | 70 | 192 | 197 | 0.29 | 0.22 | 194 | 208 | 0.59 | 0.36 | 199 | 226 | - | 0.56 |
| 0.5 | 50 | 100 | 699 | 90 | 285 | 290 | 0.19 | 0.13 | 292 | 295 | 0.38 | 0.23 | 310 | 317 | 0.95 | 0.41 |
| 1 | 50 | 100 | 770 | 90 | 288 | 295 | 0.18 | 0.15 | 292 | 301 | 0.36 | 0.26 | 303 | 331 | 0.91 | 0.48 |
| 0.5 | 50 | 10 | 134 | 90 | 229 | 233 | 0.27 | 0.16 | 233 | 234 | 0.53 | 0.23 | 242 | 251 | - | 0.35 |
| 1 | 50 | 10 | 205 | 90 | 233 | 239 | 0.25 | 0.18 | 235 | 243 | 0.51 | 0.28 | 242 | 260 | - | 0.43 |

Table 7: Same as Table 4 but for a laser exposure time of $\tau = 10$ ns.

| $p$ [bar] | $T$ [K] | $\Delta_l$ [MHz] | $\Gamma_c$ [MHz] | $\sigma_D$ [MHz] | $\mathcal{F} = 10$ J/cm$^2$ ($\Omega = 26$ MHz) $\hat{\Gamma}_V$ [MHz] | $\Gamma_V$ [MHz] | $\hat{\rho}_{33}$ | $\bar{\rho}_{33}$ | $\mathcal{F} = 20$ J/cm$^2$ ($\Omega = 36$ MHz) $\hat{\Gamma}_V$ [MHz] | $\Gamma_V$ [MHz] | $\hat{\rho}_{33}$ | $\bar{\rho}_{33}$ | $\mathcal{F} = 50$ J/cm$^2$ ($\Omega = 58$ MHz) $\hat{\Gamma}_V$ [MHz] | $\Gamma_V$ [MHz] | $\hat{\rho}_{33}$ | $\bar{\rho}_{33}$ |
|---|---|---|---|---|---|---|---|---|---|---|---|---|---|---|---|---|
| 0.5 | 22 | 100 | 770 | 60 | 245 | 257 | 0.23 | 0.14 | 269 | 258 | 0.45 | 0.24 | 328 | 291 | - | 0.42 |
| 1 | 22 | 100 | 913 | 60 | 249 | 271 | 0.21 | 0.15 | 263 | 282 | 0.41 | 0.27 | 301 | 304 | - | 0.49 |
| 0.5 | 22 | 10 | 205 | 60 | 177 | 189 | 0.36 | 0.24 | 190 | 190 | 0.72 | 0.39 | 219 | 208 | - | 0.56 |
| 1 | 22 | 10 | 348 | 60 | 184 | 205 | 0.31 | 0.22 | 194 | 210 | 0.63 | 0.38 | 218 | 227 | - | 0.60 |
| 0.5 | 30 | 100 | 733 | 70 | 271 | 276 | 0.21 | 0.11 | 301 | 278 | 0.43 | 0.20 | 371 | 312 | - | 0.34 |
| 1 | 30 | 100 | 838 | 70 | 267 | 285 | 0.20 | 0.14 | 285 | 285 | 0.40 | 0.25 | 330 | 319 | - | 0.44 |
| 0.5 | 30 | 10 | 168 | 70 | 201 | 207 | 0.33 | 0.22 | 214 | 208 | 0.65 | 0.36 | 246 | 226 | - | 0.49 |
| 1 | 30 | 10 | 273 | 70 | 203 | 215 | 0.30 | 0.21 | 214 | 221 | 0.60 | 0.35 | 240 | 238 | - | 0.54 |
| 0.5 | 50 | 100 | 693 | 90 | 333 | 314 | 0.19 | 0.08 | 374 | 315 | 0.38 | 0.13 | 470 | 351 | 0.95 | 0.22 |
| 1 | 50 | 100 | 757 | 90 | 315 | 315 | 0.18 | 0.10 | 341 | 327 | 0.37 | 0.19 | 404 | 350 | 0.91 | 0.33 |
| 0.5 | 50 | 10 | 127 | 90 | 252 | 246 | 0.27 | 0.18 | 269 | 253 | 0.53 | 0.29 | 305 | 272 | - | 0.38 |
| 1 | 50 | 10 | 192 | 90 | 249 | 252 | 0.26 | 0.18 | 263 | 258 | 0.51 | 0.30 | 293 | 285 | - | 0.43 |

to Table 4 but for a statistical distribution. Differences of less than 5% are observed for both the saturation fluences and $\bar{\rho}_{33}(\omega_r)$ at the listed experimental conditions.

Table 7 contains the transition probabilities $\bar{\rho}_{33}(\omega_r)$ and linewidths for $\tau = 10$ ns. As observed, differences in $\bar{\rho}_{33}(\omega_r)$ between 10 ns and 100 ns are more pronounced for $\Delta_l = 100$ MHz, having differences around 26%, while for $\Delta_l = 10$ MHz are negligible.

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
