# Peer review of "Laser excitation of the 1s-hyperfine transition in muonic hydrogen"

_SciPost Physics, doi:SciPost Phys. 13, 020 (2022)_

## Round 2 · Referee Report · Anonymous (Referee 1) · 2022-3-28

Strengths

This paper is extremely clearly written and carefully lays out the theoretical framework and modelling necessary to explore the feasibility of a measurement of the muonic-hydrogen ground-state hyperfine structure using the CREMA apparatus for producing muonic hydrogen, along with laser excitation, hydrogen quenching and selective detection of quenched muonic hydrogen atoms. The theoretical frame work and modelling is carefully executed and compared to simpler analytic results, where appropriate. This work is a crucial step towards a new measurement of the muonic-hydrogen ground-state hyperfine structure.

Weaknesses

None.

Report

As stated above, this work is clearly written and represents a a crucial step towards a new measurement of the muonic-hydrogen ground-state hyperfine structure. I highly recommend it for publication.

Requested changes

None.

  • validity: top
  • significance: high
  • originality: high
  • clarity: top
  • formatting: excellent
  • grammar: perfect

Author:  Pedro Amaro  on 2022-06-07  [id 2558]

(in reply to Report 1 on 2022-03-28)
Category:
remark

We are very grateful for the positive review from the referee.

---

## Round 2 · Referee Report · Anonymous (Referee 2) · 2022-4-23

Strengths

1. detailed theoretical modelling of laser spectroscopy of the 1s-hyperfine transition in muonic hydrogen (μp)
2. consider the experimental scenario as implemented by the CREMA collaboration
3. calculate matrix elements of excitation and de-excitation processes for various experimental conditions
4. show that dephasing from laser linewidth/bandwidth, elastic and inelastic collisions as well as and Doppler broadening has significant impact on excitation rates and required laser power
5. most accurate modelling of 1s-hyperfine spectroscopy in μp to date
6. well written and accessible

Weaknesses

1. claim that collisional effects are important is not unambiguously supported by the manuscript in its current form
2. some results and/or their presentation could be improved by providing a more detailed explanation/discussion

Report

The authors report on a detailed theoretical modelling of laser spectroscopy of the 1s-hyperfine transition in muonic hydrogen (μp). They consider the experimental scenario as implemented by the CREMA collaboration, in which a low-energy muon beam interacts with cold hydrogen gas to form μp, thermalized to tens of K temperature through elastic collisions with the hydrogen gas. Laser excitation to the excited hfs state is detected through the energy release upon collisional quenching of the excited μp with hydrogen, allowing the μp to leave the interaction region and to transfer the muon to a gold atom. The de-excitation of the muonic gold will result in emission of x-rays that can be detected and serve as a signal for laser excitation.
In their work they employ a simplified 3-level model for the internal states and set up optical Bloch equations to describe the dynamic evolution. They calculate matrix elements of excitation and de-excitation processes for various experimental conditions, taking into account dephasing from laser linewidth/bandwidth, elastic and inelastic collisions as well as and Doppler broadening. In comparison to previous work, the inclusion of dephasing in their model and the calculation of collision rates are new aspects, which changes some of the transition rates/saturation intensities by a factor of two.
The manuscript is well written and accessible. It presents to date the most realistic model for the planned experiments and is thus well motivated. I therefore recommend publication of this manuscript after the points below have been addressed.

Requested changes

Content
1. The authors claim that the work shows that collisional effects are “an important aspect of the experiment”. I assume that this claim is derived from Table 3, where saturation fluences in absence of decoherence and with decoherence (including collisions and laser bandwidth) are compared. This comparison does not allow to distinguish between the effect of collision and laser bandwidth. In fact, it would be very interesting, which of the two effects is dominant. From the calculated rates, it seems like collisions are dominant for a laser bandwidth of 10 MHz, whereas the laser decoherence becomes relevant for a laser bandwidth of 100 MHz. The authors should either limit their claim to what has been demonstrated or provide data to support that claim. Also, it would be interesting to learn which of the prior modelling publications ([35-38]) includes laser bandwidth.
2. Could the authors provide a figure of merit to optimize the observed signal? Is it the rate between inelastic and elastic collisions?
3. The authors assume that there is no magnetic substate-dependence of the elastic and inelastic collision rates. Is this a valid assumption? If not, this might have impact on optimal polarisation choice of the excitation radiation.
4. Table 1: Is there an intuitive picture why the elastic rate for a Boltzmann distribution is smaller, whereas the inelastic rate is larger compared to a statistical distribution?
5. Table 2: Why are the other transitions provided in the Table? The text says that this way the calculations can be confirmed. However, no comparison with other calculations/experiments except [37], which is mentioned in the footnote, is being made. Maybe add another column to compare to other work?
6. Table 3: Similarly: What is the purpose of the first two lines in the Table?
7. Figure 4: I find it curious that the population \rho_33 is smaller on resonance for higher temperatures, but the spectral shape is not broadened. Maybe the authors can comment on this.
8. Table 4, 6, 7: Why are some of the entries in the next-to-last column empty?

Typos
9. Page 3: it is not clear what is mean by “…electron-proton scattering or hydrogen [spectroscopy?]”.
10. Page 4, Section 1: “by” missing: “…results obtained [by] integrating numerically…”
11. Page 4, Section 2: singular and plural of μp atom(s) is mixed in the text
12. Page 6: The authors might want to verify the factor of 2pi when converting bandwidth to rate, since both, spontaneous emission rate and laser bandwidth, are given in FWHM in Eq. (5).
13. Page 11, Section 5: It is not clear to me what the authors mean by “Control of time-dynamics has been used to improve laser spectroscopy of molecules, muonium and highly charged ions.” Is it the fact that using Rabi or Ramsey spectroscopy gives better resolution compared to incoherent excitation?
14. Page 14, Conclusion: “undergoes” -> “undergoing”

  • validity: top
  • significance: high
  • originality: good
  • clarity: high
  • formatting: excellent
  • grammar: excellent

Author:  Pedro Amaro  on 2022-06-07  [id 2562]

(in reply to Report 2 on 2022-04-23)
Category:
remark
correction

We are thankful for the acceptance, as well as the positive criticism, which we address below:

Comment 1

  1. The authors claim that the work shows that collisional effects are “an important aspect of the experiment”. I assume that this claim is derived from Table 3, where saturation fluences in absence of decoherence and with decoherence (including collisions and laser bandwidth) are compared. This comparison does not allow to distinguish between the effect of collision and laser bandwidth. In fact, it would be very interesting, which of the two effects is dominant. From the calculated rates, it seems like collisions are dominant for a laser bandwidth of 10 MHz, whereas the laser decoherence becomes relevant for a laser bandwidth of 100 MHz. The authors should either limit their claim to what has been demonstrated or provide data to support that claim. Also, it would be interesting to learn which of the prior modelling publications ([35-38]) includes laser bandwidth.

We agree with referee, so we have modified the text in the following way:

 1. At page 4, 1rst paragraph, we replaced

“In contrast, collisional effects were considered in this work, and as we shall see, it is an important aspect of the experiment.

 with

“In contrast, decoherence effects due to collisions and laser bandwidth were considered in this work, as they reduce the transition probability by almost a factor of two at the optimal experimental conditions.”

  2. Following the referee’s comments, we highlighted that previous publications neglected decoherence effects (end of page 3 and 1rst paragraph of page 4)

“…assuming the Fermi-golden rule with Doppler convolution while neglecting both collisional effects and laser bandwidth ~\cite{Adamczak2012}.”

“…low target density~\cite{Kanda2018}}. Laser bandwidth is also omitted in this work.”

 3. As suggested by the referee we added the following sentence to help disentangle the collisional effects from laser bandwidth (end of page 10):

“…together with F_sat^(∆→10) and F_sat^(∆→100)which are the saturation fluences for ∆l=10 MHz and 100 MHz, respectively. The comparison of F_sat^(∆→10) to F_sat^(∆→100) highlights the impact of the laser bandwidth, while the comparison of F_sat^(∆→10) to F_sat^(Γ_c→0)allows to appreciate the impact of the collisional effects.”

Comment 2

2.Could the authors provide a figure of merit to optimize the observed signal? Is it the rate between inelastic and elastic collisions?

In the context of this work, it is difficult to define a figure-of-merit. The actual figure-of-merit is the statistical significance (signal/$\sqrt{\text{background}}$). The rho33 population calculated in this work is the first quantity needed to obtain and optimize this figure-of-merit. In response to the comment of the referee, we added the following sentence (page 6, end of Sec. 3.1).

“…for various target conditions and laser performances. When combined with simulations of the diffusion process, muon beam and detection system, this allows optimization of the experimental setup to maximize statistical significance (signal/$\sqrt{\text{background}}$).”

Comment 3

3.The authors assume that there is no magnetic substate-dependence of the elastic and inelastic collision rates. Is this a valid assumption? If not, this might have impact on optimal polarisation choice of the excitation radiation.

The rate of mu-p + p scattering depends on the absolute value of total spin S of the two colliding particles. On the other hand, this rate does not depend on the projection of vector S on a fixed axis (magnetic level m). This can be found in publications about this topic as in L. Bracci et al, Phys. Lett. A 134, 435 (1989), L. Bracci et al, Phys. Lett. A 149, 463 (1990),

We agree with the referee that this is an important detail. Hence, we added the following sentence (end of page 6) and two references:

“Note that these collisional cross-sections do not depend on the magnetic sub-states of the $\mu$p atom \cite{Bracci1989, Bracci1990}.”

Comment 4

4.Table 1: Is there an intuitive picture why the elastic rate for a Boltzmann distribution is smaller, whereas the inelastic rate is larger compared to a statistical distribution?

There is no intuitive picture for this observation. In the following we provide an explanation for the referee, but we refrain adding it in the main text as it is a complex explanation and beyond the scope of this work.

In the scattering of mu-p atoms from the H_2 molecules, collisions take place on a single proton bound in the H_2. The corresponding scattering amplitudes strongly depend on the total spin S of the mu-p + p system. The elastic scattering can take place in the two states S = 1/2 and S = 3/2, with the magnitudes of scattering amplitudes differing more than one order of magnitude. On the other hand, the inelastic scattering is possible only for S=1/2. Since the H_2 molecule is symmetric, the even and odd rotational states are characterized by different total nuclear spins of 0 and 1, respectively. As a result, the mu-p collisions with H_2 in the ortho and para states are connected with specific spin statistical factors, which are different for the elastic and inelastic scattering. Since the population of the Boltzmann distribution at a low temperature is dominated by the ground rotational state and the statistical distribution has a 75%-contribution from the first odd rotational state, the values of the elastic and inelastic rates independently vary between the two distributions.

Comment 5

5.Table 2: Why are the other transitions provided in the Table? The text says that this way the calculations can be confirmed. However, no comparison with other calculations/experiments except [37], which is mentioned in the footnote, is being made. Maybe add another column to compare to other work?

The 2s-2p transitions are provided to highlight that these matrix elements in muonic atoms are very small due to small size of these atoms. The Bohr radius scales inversely with the orbiting particle mass. In addition, it allows us to emphasize the two orders of magnitude difference in the hyperfine splitting. To make this clearer in the text we made the following modifications in Table 2 caption:

“The analytical expressions for the $2s-2p$ matrix elements agree with \cite{Schmidt2018}. $a_\mu$ is the muonic Bohr radius.”

Comment 6

6.Table 3: Similarly: What is the purpose of the first two lines in the Table?

The purpose is to highlight some fundamental differences (widths, lifetimes, etc…) with respect to previous experiment. Moreover, it highlights the challenge faced by the new experiment that needs to provide four orders of magnitude higher fluence.

Following the remark of the referee, we specified this last aspect in the following sentence (end of page 10)

"…transitions in $\mu$p, emphasizing the laser technology leap needed to accomplish the hyperfine experiment."

Comment 7

7.Figure 4: I find it curious that the population \rho_33 is smaller on resonance for higher temperatures, but the spectral shape is not broadened. Maybe the authors can comment on this.

The broadening is not clearly visible in this figure because the various resonances have different amplitudes. To avoid this possible confusion pointed out by the referee we added the following sentence at the end of the Figure 4 caption.

"Values of \rho_33 on resonance and FWHM linewidth can be found in Table 4. "

Comment 8

8.Table 4, 6, 7: Why are some of the entries in the next-to-last column empty?

This column corresponds to the analytical approximation expressed by eq. (18), valid for low fluences. For high fluences, this expression can give values that exceed a probability of one. For this reason, we do not quote these values as they are non-physical.

To avoid confusion, we add the following sentence to the end of the table 4 caption.

“The symbol “–“ indicates that \rho_33>1, which is non-physical.”

Comment 9

9.Page 3: it is not clear what is mean by “…electron-proton scattering or hydrogen [spectroscopy?]”.

Thank for nothing this. We added the term “spectroscopy”.

Comment 10

10.Page 4, Section 1: “by” missing: “…results obtained [by] integrating numerically…”

We corrected.

Comment 11

11.Page 4, Section 2: singular and plural of μp atom(s) is mixed in the text

We corrected.

Comment 12

12.Page 6: The authors might want to verify the factor of 2pi when converting bandwidth to rate, since both, spontaneous emission rate and laser bandwidth, are given in FWHM in Eq. (5).

The factor of 2pi is correct. It has been verified independently by three authors independently.

Comment 13

13.Page 11, Section 5: It is not clear to me what the authors mean by “Control of time-dynamics has been used to improve laser spectroscopy of molecules, muonium and highly charged ions.” Is it the fact that using Rabi or Ramsey spectroscopy gives better resolution compared to incoherent excitation?

The referee is correct that the sentence is confusing. We modified it in the following way:

“Simulating the time-evolution of state populations using Bloch equations has improved laser spectroscopy of molecules [], muonium [] and highly charged ions []. Similarly, the findings of population dynamics obtained from Eq. (1)-(4) can be used to optimize the target conditions (T and p) of the HFS experiment.”

Comment 14

14.Page 14, Conclusion: “undergoes” -> “undergoing”

To avoid possible grammar confusion, we simplified the sentence:

“…probability that a thermalized μp atom undergoes laser excitation from the singlet to triplet states …”

---

## Round 2 · Referee Report · Anonymous (Referee 3) · 2022-5-11

Strengths

  1. Careful and detailed theoretical analysis of the proposed CREMA experiment to measure the ground-state hyperfine splitting in muonic hydrogen.
  2. Expected high significance of the results in terms of the proton size and fundamental physics.

Weaknesses

  1. The paper should be carefully proof-read. For example, after Eq. (11) in the text, M1 should presumably be E1 for the 2s -> 2p transition. also, the authors' names are repeated in Ref. 1.

Report

The journal's acceptance criteria are met, and the paper should be accepted for publication after careful proof-reading and correction of minor errors as noted under "weaknesses."

Requested changes

  1. Correct the minor errors as noted under "weaknesses."

  • validity: high
  • significance: high
  • originality: high
  • clarity: high
  • formatting: excellent
  • grammar: good

Author:  Pedro Amaro  on 2022-06-07  [id 2560]

(in reply to Report 3 on 2022-05-11)

An English native-speaker has proof-read the paper.

We corrected the typo pointed out by the referee about the first reference having duplications.

Moreover, references 10 and 28 have been updated.
We homogenized the DOI style display in all references.

We agree that the sentence was unclear. We therefore simplified the sentence as:

“The matrix elements $\mathcal{M}^{(m, m')}$ for the $2s-2p$ (E1-type) and 1s-HFS (M1-type) transitions are given by”

---

## Round 3 · Referee Report · Anonymous (Referee 3) · 2022-6-25

Report

The authors have satisfactorily answered all my questions.

---

## Round 3 · Referee Report · Anonymous (Referee 2) · 2022-6-26

Report

This revised manuscript is ready for acceptance and publication.

---

## Round 3 · Referee Report · Anonymous (Referee 1) · 2022-6-28

Report

The authors have satisfactorily addressed the previous comments of the referees, and the paper is now ready to be accepted for publication.

---

## Round 3 · Author Response

Dear Editor and Referees,

Thank you very much for the constructive review. We have carefully addressed your critiques and correspondingly modified the manuscript, which we here resubmit for publication in SciPost Physics.

Besides the ArXiv version of the manuscript (2112.00138v3) that we resubmitted with all the suggested modifications (see below), we also send by email to you another version (“paper_HFS_calcu_v11_sim_resu.pdf”) with all subsequent modifications made to the paper displayed in blue, to make them clearer. We would appreciate if you could forward this file to the referees.

---

## Round 3 · List of Changes

We are thankful for the acceptance, as well as the positive criticism, which we address below:

---->Anonymous Report 2 on 2022-4-23

Comment 1
1. The authors claim that the work shows that collisional effects are “an important aspect of the experiment”. I assume that this claim is derived from Table 3, where saturation fluences in absence of decoherence and with decoherence (including collisions and laser bandwidth) are compared. This comparison does not allow to distinguish between the effect of collision and laser bandwidth. In fact, it would be very interesting, which of the two effects is dominant. From the calculated rates, it seems like collisions are dominant for a laser bandwidth of 10 MHz, whereas the laser decoherence becomes relevant for a laser bandwidth of 100 MHz. The authors should either limit their claim to what has been demonstrated or provide data to support that claim. Also, it would be interesting to learn which of the prior modelling publications ([35-38]) includes laser bandwidth.

We agree with referee, so we have modified the text in the following way:

At page 4, 1rst paragraph, we replaced

“In contrast, collisional effects were considered in this work, and as we shall see, it is an important aspect of the experiment.

with

“In contrast, decoherence effects due to collisions and laser bandwidth were considered in this work, as they reduce the transition probability by almost a factor of two at the optimal experimental conditions.”

Following the referee’s comments, we highlighted that previous publications neglected decoherence effects (end of page 3 and 1rst paragraph of page 4)

“…assuming the Fermi-golden rule with Doppler convolution while neglecting both collisional effects and laser bandwidth ~\cite{Adamczak2012}.”

“…low target density~\cite{Kanda2018}}. Laser bandwidth is also omitted in this work.”

As suggested by the referee we added the following sentence to help disentangle the collisional effects from laser bandwidth (end of page 10):

“…together with F_sat^(∆→10) and F_sat^(∆→100)which are the saturation fluences for ∆l=10 MHz and 100 MHz, respectively. The comparison of F_sat^(∆→10) to F_sat^(∆→100) highlights the impact of the laser bandwidth, while the comparison of F_sat^(∆→10) to F_sat^(Γ_c→0)allows to appreciate the impact of the collisional effects.”

Comment 2
2. Could the authors provide a figure of merit to optimize the observed signal? Is it the rate between inelastic and elastic collisions?

In the context of this work, it is difficult to define a figure-of-merit. The actual figure-of-merit is the statistical significance (signal/sqrt(background)). The rho33 population calculated in this work is the first quantity needed to obtain and optimize this figure-of-merit. In response to the comment of the referee, we added the following sentence (page 6, end of Sec. 3.1).

“…for various target conditions and laser performances. When combined with simulations of the diffusion process, muon beam and detection system, this allows optimization of the experimental setup to maximize statistical significance (signal/$\sqrt{\text{background}}$).”

Comments 3

3. The authors assume that there is no magnetic substate-dependence of the elastic and inelastic collision rates. Is this a valid assumption? If not, this might have impact on optimal polarisation choice of the excitation radiation.

The rate of mu-p + p scattering depends on the absolute value of total spin S of the two colliding particles. On the other hand, this rate does not depend on the projection of vector S on a fixed axis (magnetic level m). This can be found in publications about this topic as in
L. Bracci et al, Phys. Lett. A 134, 435 (1989),
L. Bracci et al, Phys. Lett. A 149, 463 (1990),

We agree with the referee that this is an important detail. Hence, we added the following sentence (end of page 6) and two references:

“Note that these collisional cross-sections do not depend on the magnetic sub-states of the $\up$ atom \cite{Bracci1989, Bracci1990}.”

Comment 4
4. Table 1: Is there an intuitive picture why the elastic rate for a Boltzmann distribution is smaller, whereas the inelastic rate is larger compared to a statistical distribution?

There is no intuitive picture for this observation. In the following we provide an explanation for the referee, but we refrain adding it in the main text as it is a complex explanation and beyond the scope of this work.

In the scattering of mu-p atoms from the H_2 molecules, collisions take place on a single proton bound in the H_2. The corresponding scattering amplitudes strongly depend on the total spin S of the mu-p + p system. The elastic scattering can take place in the two states S = 1/2 and S = 3/2, with the magnitudes of scattering amplitudes differing more than one order of magnitude. On the other hand, the inelastic scattering is possible only for S=1/2. Since the H_2 molecule is symmetric, the even and odd rotational states are characterized by different total nuclear spins of 0 and 1, respectively. As a result, the mu-p collisions with H_2 in the ortho and para states are connected with specific spin statistical factors, which are different for the elastic and inelastic scattering. Since the population of the Boltzmann distribution at a low temperature is dominated by the ground rotational state and the statistical distribution has a 75%-contribution from the first odd rotational state, the values of the elastic and inelastic rates independently vary between the two distributions.

Comment 5
5. Table 2: Why are the other transitions provided in the Table? The text says that this way the calculations can be confirmed. However, no comparison with other calculations/experiments except [37], which is mentioned in the footnote, is being made. Maybe add another column to compare to other work?

The 2s-2p transitions are provided to highlight that these matrix elements in muonic atoms are very small due to small size of these atoms. The Bohr radius scales inversely with the orbiting particle mass. In addition, it allows us to emphasize the two orders of magnitude difference in the hyperfine splitting. To make this clearer in the text we made the following modifications in Table 2 caption:

“The analytical expressions for the $2s-2p$ matrix elements agree with \cite{Schmidt2018}. $a_\mu$ is the muonic Bohr radius.”

Comment 6
6. Table 3: Similarly: What is the purpose of the first two lines in the Table?

The purpose is to highlight some fundamental differences (widths, lifetimes, etc…) with respect to previous experiment. Moreover, it highlights the challenge faced by the new experiment that needs to provide four orders of magnitude higher fluence.

Following the remark of the referee, we specified this last aspect in the following sentence (end of page 10)

“…transitions in $\mu$p, emphasizing the laser technology leap needed to accomplish the hyperfine experiment.}”

Comment 7
7. Figure 4: I find it curious that the population \rho_33 is smaller on resonance for higher temperatures, but the spectral shape is not broadened. Maybe the authors can comment on this.

The broadening is not clearly visible in this figure because the various resonances have different amplitudes. To avoid this possible confusion pointed out by the referee we added the following sentence at the end of the Figure 4 caption.

“Values of \rho_33 on resonance and FWHM linewidth can be found in Table 4.”

Comment 8
8. Table 4, 6, 7: Why are some of the entries in the next-to-last column empty?

This column corresponds to the analytical approximation expressed by eq. (18), valid for low fluences. For high fluences, this expression can give values that exceed a probability of one. For this reason, we do not quote these values as they are non-physical.

To avoid confusion, we add the following sentence to the end of the table 4 caption.

“The symbol “–“ indicates that \rho_33>1, which is non-physical.”

Comment 9
9. Page 3: it is not clear what is mean by “…electron-proton scattering or hydrogen [spectroscopy?]”.

Thank for nothing this. We added the term “spectroscopy”.

Comment 10
10. Page 4, Section 1: “by” missing: “…results obtained [by] integrating numerically…”
We corrected.

Comment 11
11. Page 4, Section 2: singular and plural of μp atom(s) is mixed in the text
We corrected.

Comment 12
12. Page 6: The authors might want to verify the factor of 2pi when converting bandwidth to rate, since both, spontaneous emission rate and laser bandwidth, are given in FWHM in Eq. (5).

The factor of 2pi is correct. It has been verified independently by three authors independently.

Comment 13
13. Page 11, Section 5: It is not clear to me what the authors mean by “Control of time-dynamics has been used to improve laser spectroscopy of molecules, muonium and highly charged ions.” Is it the fact that using Rabi or Ramsey spectroscopy gives better resolution compared to incoherent excitation?

The referee is correct that the sentence is confusing. We modified it in the following way:

“Simulating the time-evolution of state populations using Bloch equations has improved laser spectroscopy of molecules [], muonium [] and highly charged ions []. Similarly, the findings of population dynamics obtained from Eq. (1)-(4) can be used to optimize the target conditions (T and p) of the HFS experiment.”

Comment 14
14. Page 14, Conclusion: “undergoes” -> “undergoing”
To avoid possible grammar confusion, we simplified the sentence:

“…probability that a thermalized μp atom undergoes laser excitation from the singlet to triplet states …”

---->Anonymous Report 3 on 2022-5-11

The paper should be carefully proof-read. For example, after Eq. (11) in the text, M1 should presumably be E1 for the 2s -> 2p transition. also, the authors' names are repeated in Ref. 1.

An English native-speaker has proof-read the paper.

We corrected the typo pointed out by the referee about the first reference having duplications.

Moreover, references 10 and 28 have been updated.
We homogenized the DOI style display in all references.

We agree that the sentence was unclear. We, therefore, simplified the sentence as:

“The matrix elements $\mathcal{M}^{(m, m')}$ for the $2s-2p$ (E1-type) and 1s-HFS (M1-type) transitions are given by”

---

## Editorial Decision

published